# Obesity and smoking as risk factors for invasive mechanical ventilation in COVID-19: A retrospective, observational cohort study

Ana C. Monteiro[1]*, Rajat Suri[1], Iheanacho O. Emeruwa[1], Robert J. Stretch[1], Roxana Y. Cortes-Lopez[1], Alexander Sherman[1], Catherine C. Lindsay[2], Jennifer A. Fulcher[3], David Goodman-Meza[3], Anil Sapru[4], Russell G. Buhr[1], Steven Y. Chang[1], Tisha Wang[1]©, Nida Qadir[1]©

1 Division of Pulmonary and Critical Care, Department of Medicine, UCLA Medical Center, Los Angeles, CA, United States of America, 2 Department of Medicine, UCLA Medical Center, Los Angeles, CA, United States of America, 3 Division of Infectious Disease, Department of Medicine, UCLA Medical Center, Los Angeles, CA, United States of America, 4 Division of Critical Care, Department of Pediatrics, UCLA Medical Center, Los Angeles, CA, United States of America

© These authors contributed equally to this work.
* acostamonteiro@mednet.ucla.edu

**Data Availability Statement:** The data are available at DOI: https://doi.org/10.5068/D1QX18.

## Abstract

### Purpose

To describe the trajectory of respiratory failure in COVID-19 and explore factors associated with risk of invasive mechanical ventilation (IMV).

### Materials and methods

A retrospective, observational cohort study of 112 inpatient adults diagnosed with COVID-19 between March 12 and April 16, 2020. Data were manually extracted from electronic medical records. Multivariable and Univariable regression were used to evaluate association between baseline characteristics, initial serum markers and the outcome of IMV.

### Results

Our cohort had median age of 61 (IQR 45–74) and was 66% male. In-hospital mortality was 6% (7/112). ICU mortality was 12.8% (6/47), and 18% (5/28) for those requiring IMV. Obesity (OR 5.82, CI 1.74–19.48), former (OR 8.06, CI 1.51–43.06) and current smoking status (OR 10.33, CI 1.43–74.67) were associated with IMV after adjusting for age, sex, and high prevalence comorbidities by multivariable analysis. Initial absolute lymphocyte count (OR 0.33, CI 0.11–0.96), procalcitonin (OR 1.27, CI 1.02–1.57), IL-6 (OR 1.17, CI 1.03–1.33), ferritin (OR 1.05, CI 1.005–1.11), LDH (OR 1.57, 95% CI 1.13–2.17) and CRP (OR 1.13, CI 1.06–1.21), were associated with IMV by univariate analysis.

### Conclusions

Obesity, smoking history, and elevated inflammatory markers were associated with increased need for IMV in patients with COVID-19.

**Funding:** ACM is funded by the National Institutes of Health (NIH) training grant T32 5T32HL072752-1 The funders had no role in study design, data collection and analysis, decision to publish, or preparation of the manuscript.

**Competing interests:** I have read the journal's policy and the authors of this manuscript have the following competing interests: SYC consults for PureTech on their deupirfenidone in COVID study. This does not alter our adherence to PLOS ONE policies on sharing data and materials.

## Introduction

Coronavirus disease 2019 (COVID-19) has been reported in over 200 countries [1], leading to an unprecedented impact on healthcare systems worldwide. While its disease course is incompletely understood, its causative virus, SARS-CoV-2, is thought to enter respiratory epithelial cells via the angiotensin-converting enzyme-2 (ACE2) receptor in the lungs, resulting in a variety of clinical symptoms. Given the high infection rates worldwide, the identification of markers predicting increased disease severity may enable more effective triaging of patients and procurement of resources.

Due to its novelty and heterogeneity in presentation, the underlying mechanism for the severe hypoxic and hypercapnic respiratory failure sometimes seen in COVID-19 has been the subject of controversy. The common features of bilateral pulmonary infiltrates in combination with low PaO2/FiO2 (P/F) ratios are consistent with acute respiratory distress syndrome (ARDS). However, other potential etiologies for COVID-19-related respiratory failure have been proposed, including an atypical, "high compliance" form of ARDS, high-altitude pulmonary edema, and other non-pulmonary causes of hypoxia, such as those caused by cardiomyopathy [2–5]. The presence of various hypotheses for disease mechanism may be playing a role in the management and outcomes of patients with respiratory failure.

We present our cohort of the first 112 unique, consecutively admitted patients to our hospital system with confirmed COVID-19 infection. Our aim was to describe characteristics, management, and trajectory of respiratory failure and mortality in our cohort, and to explore the factors associated with the need for invasive mechanical ventilation.

## Methods

### Setting, patient population, and study design

This is a retrospective observational cohort study conducted from March 12 2020 and April 16 2020 approved by the UCLA (University of California, Los Angeles) institutional review board with waiver of informed consent. We evaluated the first 113 unique admissions to our healthcare system. We originally accessed patient charts between March 12 and June 16th 2020. We returned to the charts in October 2020 to extract elements requested during the review process. The UCLA hospital system is an academic center in Los Angeles County comprised of Ronald Reagan-UCLA Medical Center (RR-UCLA) and Santa Monica-UCLA Medical Center (SM-UCLA). RR-UCLA has 520 beds, of which 109 are critical care beds, while SM-UCLA has 281 beds, of which 22 are ICU beds. Hospitalized patients at RR-UCLA and SM-UCLA ≥ 18 years old with positive SARS-CoV-2 PCR testing from either nasal swab or mini-bronchoalveolar lavage (BAL) testing were included. Data for patients who tested positive for SARS-CoV-2 was manually extracted from the electronic health record and included in a database.

### Clinical protocols

This was an observational study, and all clinical management was left to the discretion of the primary treatment team. A pandemic response team comprised of intensivists and infectious disease specialists generated COVID-19 treatment guidance documents; recommendations emphasized established best critical care practices. Guidelines recommended intubation for patients who had PCR confirmed COVID-19 and who demonstrated rapid escalation of oxygen requirements. It was at the discretion of the clinician to decide whether a patient would tolerate a trial of non-invasive oxygen delivery via non-rebreather mask or high flow nasal canula (HFNC). If the clinician decided that trial of these non-invasive therapies would place the patient and/or staff at greater risk, the patient would proceed to intubation.

Recommendations for patients who developed acute respiratory distress syndrome (ARDS) included low tidal volume ventilation with tidal volumes ≤ 6 mL/kg predicted body weight (PBW), goal plateau pressures under 30 cm $H_2O$ when possible, early consideration of prone ventilation for patients with P/F ratios < 150, daily awakening trials or light sedation, and conservative fluid management. Titration of positive end-expiratory pressure (PEEP) was recommended to reflect ARDS network PEEP tables [6–9].

All admitted patients with COVID-19 were considered for enrollment in interventional clinical trials. Clinical trials available at our centers from March 12, 2020 to April 16, 2020 included those testing the therapeutic benefit of sarilumab (NCT04315298), remdesivir (NCT 04280705), leronlimab (NCT04347239) and hydroxychloroquine (NCT04332991). In addition, during this time, several patients received open-label remdesivir, leronlimab, hydroxychloroquine and tocilizumab (S1 and S2 Tables in S1 File).

## Patient and public involvement

This was a de-identified chart review and as such no patients were directly involved.

## Data collection

Baseline demographic variables, smoking history, comorbidities, and body mass index (BMI) were collected. Inflammatory markers including C-reactive protein (CRP), D-dimer, ferritin, lactate dehydrogenase (LDH) and highly-sensitive Interleukin-6 (IL-6); and other labs such as hemoglobin A1c, troponin, procalcitonin and a white blood cell count with differential were obtained on admission, or on day of COVID-19 diagnosis if the cause of the original admission was unrelated to confirmed or suspected COVID-19. Treatment-level variables collected included use of anticoagulation, enrollment in clinical trials, as well as the use of open-label directed therapies for COVID-19, which included hydroxychloroquine, tocilizimab, leronlimab, and remdesivir. For those patients on mechanical ventilation, ventilator and respiratory parameters were collected, as were the use of adjunctive ARDS therapies, including prone positioning, neuromuscular blockade, pulmonary vasodilators, and extracorporeal support. Outcomes data that were collected included 60 day in-hospital mortality, ICU admission, rate of endotracheal intubation, ICU length of say, hospital length of stay, and duration of mechanical ventilation. We defined adherence to lung protective ventilation as the use of tidal volume ($V_T$) < 8 mL/kg predicted body weight (PBW).

## Definitions

We identified patients who had ARDS as defined by the Berlin Criteria—P/F ratio < 300 with chest imaging revealing bilateral opacities that could not be exclusively explained by cardiogenic causes [10]. We extracted smoking status from the admission note. Patients were interviewed about smoking history on admission. Never smokers, prior smokers and current smokers were self-identified as such by the patients during the interview. Those patients who self-identified as prior smokers were then asked when they last smoked and number of packs a day. Those who identified as current smokers were asked the number of packs smoked a day. Patients who smoked less than a pack-year were considered never smokers. Diagnosis of venous thromboembolism (VTE) was considered present if there was documented radiographic evidence of either pulmonary embolism (via computerized tomography) or deep venous thrombosis (via ultrasound). We defined mortality as death during the first 60 days of inpatient hospitalization. Any admission to the ICU or any intubation for respiratory failure, regardless of duration, was included in the rate of ICU admissions and intubations, respectively.

## Statistical methods

We combined key categorical variables so that smoking history was evaluated as present, former, or never-smoker, and comorbidities as no known medical history versus any past medical history. We also evaluated these categorical variables in all their individual categories when providing descriptive statistics. Race/ethnicity was defined as White, Black, Latinx, Asian, or other. Descriptive statistics employing simple mean, median and interquartile range were used for the baseline characteristics of the whole cohort and by intubation status. Two-by-two tables were used to describe outcomes of selected subgroups. Chi-squared analysis was used to assess statistical significance between categorical variables, two sample t-test and two sample Wilcoxon Rank-Sum tests were used to assess statistical significance between continuous variables. Wilcoxon Rank-Sum test was used when the total number was low and the assumption of a normal distribution was not met.

Multivariable logistic regression was utilized to assess the contribution of selected baseline characteristics of the cohort to the odds of the binary outcome of requiring mechanical ventilation. Past medical history with cohort prevalence of >15% were used in the multivariable model. We also explored association between selected pro-inflammatory markers and odds of requiring mechanical ventilation by using univariate logistic regression models for each biomarker. Multivariable analysis was not used for biomarkers because of significant co-linearity between the inflammatory markers. We established a threshold of 15% missingness for key variables of interest *a priori* to trigger multiple imputation, which was not met for any variable in our analyses.

## Results

### Patient characteristics

We evaluated the first 113 unique admissions to our healthcare system with confirmed COVID-19 infection, as diagnosed between March 12, 2020 and April 16, 2020. We excluded one patient who incidentally tested positive for COVID-19 but died from complications from a motor vehicle collision before COVID-directed inpatient management was initiated.

Among the remaining 112 patients, the median age was 61 years old (IQR 45–74) and subjects were predominantly male (66%). The cohort was 44% White, 29% Latinx, 8% Asian, 6% Black, and 13% other. The majority (84%) had a known comorbidity on admission, with the most frequent comorbidities being diabetes (65%), hypertension (50%), obesity (36%), chronic kidney disease (17%), and coronary artery disease (15%) (Table 1). Of our cohort, 27% were taking either ACE-inhibitors or angiotensin receptor blockers (ARBs) as an outpatient, and 7% were health care workers. Four patients were admitted as a transfer from an outside hospital for higher level of care. Out of the 47 ICU admissions, the median APACHE II score on arrival to the ICU was 12 (IQR 7–16, mean 12.7, SD 7.11).

### Rate of admissions

As of April 17, the day after the last COVID diagnosis for this cohort, there were 11,391 confirmed COVID-19 cases reported in Los Angeles County. During this period, there were 113 unique COVID related admissions in our hospital system at a rate of 1–8 patients per day. The peak number of new COVID-related admissions was 8 on April 3rd (S1 Fig in S1 File).

### Inflammatory markers on admission

Our cohort had a notable elevation of pro-inflammatory markers on presentation without an elevated white blood count (median 6.25 x $10^3$/μL IQR 4.75–8.55 x $10^3$/μL). Median

**Table 1. Baseline characteristics of cohort.**

| | | Total (N = 112) | Never Intubated (N = 84) | Intubated (N = 28) | P-value |
|---|---|---|---|---|---|
| **Sex** | Male | 74 (66%) | 55 (65%) | 19 (68%) | 0.818 |
| | Female | 38 (34%) | 29 (35%) | 9 (32%) | |
| **Age** | | 61 (45–74) | 64 (43–77) | 58 (51–64) | 0.320 |
| **Race/Ethnicity** | White | 49 (44%) | 41 (49%) | 8 (29%) | 0.350 |
| | Latinx | 33 (29%) | 24 (29%) | 9 (32%) | |
| | Asian | 9 (8%) | 6 (7%) | 3 (11%) | |
| | Black | 7 (6%) | 4 (5%) | 3 (11%) | |
| | Other | 14 (13%) | 9 (11%) | 5 (18%) | |
| **Past Medical History** | No PMHx | 17 (15%) | 14 (17%) | 3 (11%) | 0.447 |
| | Obesity | 40 (36%) | 23 (27%) | 17 (61%) | 0.001 |
| | Hypertension | 56 (50%) | 39 (46%) | 17 (61%) | 0.190 |
| | Diabetes | 72 (64%) | 53 (63%) | 19 (68%) | 0.649 |
| | COPD | 6 (5.4%) | 4 (5%) | 2 (7%) | 0.628 |
| | CAD | 17 (15%) | 14 (17%) | 3 (11%) | 0.447 |
| | Cancer | 15 (13%) | 13 (15%) | 2 (7%) | 0.262 |
| | Asthma | 13 (12%) | 9 (11%) | 4 (14%) | 0.609 |
| | Atrial Fibrillation | 11 (10%) | 10 (12%) | 1 (4%) | 0.199 |
| | CKD | 19 (17%) | 16 (19%) | 3 (11%) | 0.309 |
| | Transplant Recipient | 7 (6%) | 6 (7%) | 1 (4%) | 0.499 |
| **Tobacco Exposure History** | Never Smoker | 77 (69%) | 63 (75%) | 14 (50%) | 0.034 |
| | Former Smoker | 20 (18%) | 14 (17%) | 6 (21%) | |
| | Current Smoker | 7 (6%) | 3 (4%) | 4 (14%) | |
| | Unknown | 8 (7%) | 4 (5%) | 4 (14%) | |

All data reported as n(%) with exception of age which is reported as Median (IQR). reported p-value is from Chi Squared test with exception of age where a two-sided t-test was used. PMHx: Past Medical History; COPD: Chronic Obstructive Pulmonary Disease; CAD: Coronary Artery Disease; CKD: Chronic Kidney Disease.

interleukin-6 level was 9 pg/ml on admission (upper limit of normal for assay <5 pg/mL) (IQR 2–23 pg/ml), median D-dimer was 1140 ng/ml (IQR 677–2073 ng/ml), and median ferritin level was 696 ng/ml (IQR = 357–1616 ng/ml). The median absolute lymphocyte count was 0.9 x $10^3$/μL, (IQR 0.58–1.18 x $10^3$/μL), median CRP was 7.8 mg/dl (IQR 3.2–12.65 mg/dl), median LDH was 324 U/L (IQR 239–423), and median procalcitonin was 0.12 ug/L (IQR 0–0.385 ug/L) (Table 2).

**Table 2. Univariate logistic regression model for requiring mechanical ventilation.**

| | N | Median (IQR) | Odds Ratio (95% CI) | P-value |
|---|---|---|---|---|
| **Absolute Lymphocyte Count (10^3/uL)** | 111 | 6.25 (4.8–8.5) | 0.33 (0.11–0.96) | 0.042 |
| **LDH (U/L)**[*] | 102 | 324 (239–423) | 1.57 (1.13–2.17) | 0.006 |
| **Ferritin (ng/ml)**[*] | 98 | 696 (357–1616) | 1.05 (1.005–1.11) | 0.032 |
| **C-Reactive Protein (mg/dl)** | 100 | 7.8 (3.2–12.65) | 1.13 (1.06–1.21) | <0.001 |
| **D-Dimer (ng/ml)**[*] | 99 | 1140 (677–2073) | 1.02 (0.99–1.04) | 0.053 |
| **Procalcitonin (μg/L)** | 108 | 0.12 (0–0.385) | 1.27 (1.02–1.57) | 0.030 |
| **Interleukin-6 (pg/ml)**[**] | 91 | 9 (2–23) | 1.17 (1.03–1.33) | 0.015 |

[*]per 100-unit change of inflammatory marker.

[**]per 10-unit change of inflammatory marker.

## Patient outcomes

Of the 112 patients evaluated, there were 47 ICU admissions, 28 intubations and 7 deaths. Of the 112 patients, all had completed disposition at the time of this report. The rate of in-hospital 60-day mortality for our total cohort was 6% (7/112) with an ICU mortality of 12.8% (6/47), and a mortality of 18% (5/28) for those who required endotracheal intubation. Two patients received CPR, neither of whom achieved return of spontaneous circulation. Of the 7 patients who died, 2 had a do not resuscitate (DNR) order documented in the chart at the time of admission. Median ICU length of stay (LOS) was 7 days (IQR 3–15 days).

## Course of respiratory failure

We evaluated individual patients' trajectories of respiratory failure. High-flow nasal cannula (HFNC) was required for 11 patients, of whom 45% progressed to intubation. Conversely, of those who were intubated, only 18% were previously on HFNC. Baseline factors associated with need for intubation were obesity (OR 5.82, p = 0.004), former smoker status (OR 8.06, p = 0.02), and current smoking status (OR 10.33, p = 0.02), as per multivariable logistic regression analysis adjusting for age, sex, and high prevalence comorbidities (Table 3 and Fig 1). Additionally, the proinflammatory markers IL-6, ferritin, LDH and CRP, drawn on admission or on day of COVID-19 diagnosis for patients already hospitalized at the time of positive PCR, were associated with higher odds of intubation by univariate analysis (Table 3). The absolute lymphocyte count was also associated with need for mechanical ventilation (OR of 0.33, CI 0.11–0.96).

Out of the 28 patients intubated, 24 patients were diagnosed with ARDS (21% of our total cohort), and 21 patients had a P/F ratio < 150 (19% of our total cohort). An additional three patients had bilateral opacities and required HFNC. The ICU LOS was significantly greater amongst patients with ARDS. Those with ARDS had an ICU LOS of 15 days (IQR 9–24 days) while those without ARDS had an ICU LOS of 5 days (IQR 2–5 days; difference with p< 0.01).

**Table 3. Multivariable logistic regression for odds of requiring mechanical ventilation.**

|  |  | Odds Ratio (95% CI) | P-value |
|---|---|---|---|
| **Sex** | Female | Reference |  |
|  | Male | 0.58 (0.18–1.92) | 0.38 |
| **Race/Ethnicity** | White | Reference |  |
|  | Black | 5.19(0.81–33.44) | 0.08 |
|  | Latinx | 2.63(0.63–11.03) | 0.19 |
|  | Asian | 5.44(0.62–47.71) | 0.13 |
|  | Other | 2.98 (0.46–19.26) | 0.25 |
| **Age (per 1 year increase)** |  | 0.99 (0.96–1.03) | 0.76 |
| **Past Medical History** | Obesity | 5.82(1.74–19.48) | <0.01 |
|  | Diabetes | 1.71 (0.55–5.37) | 0.36 |
|  | Hypertension | 2.28 (0.68–7.61) | 0.18 |
|  | CAD | 0.48 (0.08–3.08) | 0.44 |
|  | CKD | 0.20 (0.03–1.15) | 0.07 |
| **Tobacco exposure history** | Never Smoker | Reference |  |
|  | Former Smoker | 8.06 (1.51–43.06) | 0.02 |
|  | Current Smoker | 10.33 (1.43–74.67) | 0.02 |
|  | Unknown | 8.63 (1.06–70.12) | 0.04 |

Comorbidities with cohort prevalence of greater than 15 percent were included in this regression. CAD: Coronary Artery Disease; CKD: Chronic Kidney Disease.

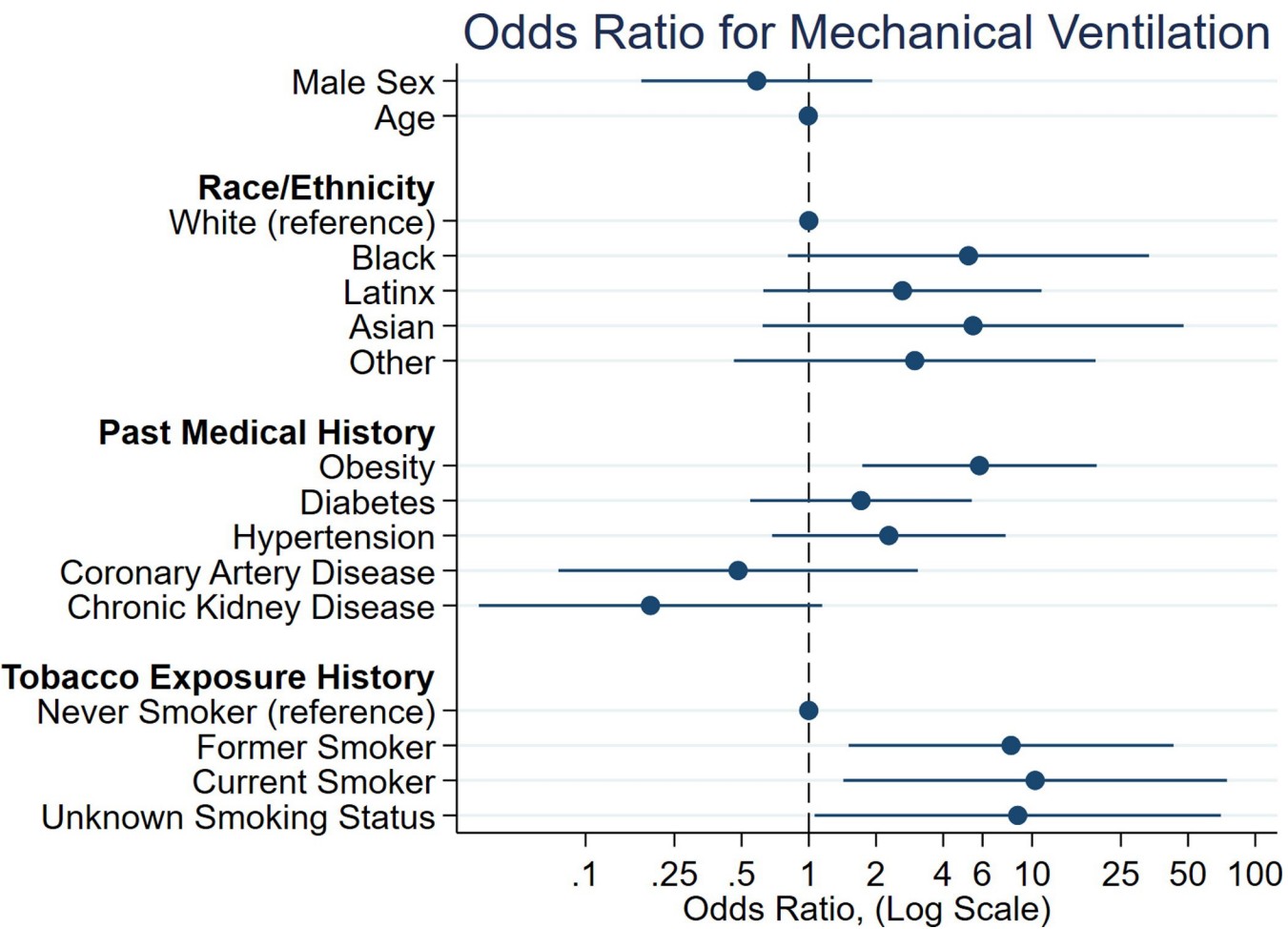

**Fig 1. Forest plot of multivariable logistic regression analysis adjusting for age, sex, and comorbidities with cohort prevalence of ≥15%.** The x-axis is depicted on a log scale.

### Pharmacologic therapies

During the period of data collection, there was no definitive evidence supporting or discrediting the use of selected anti-inflammatory or anti-viral medications for the treatment of COVID-19. In our cohort, 54% of patients were either enrolled in a placebo-controlled trial or received open label use of one or more of the following therapies: anti-IL-6 therapy (sarilumab or tocilizumab), CCR5 antagonist (leronlimab), hydroxychloroquine, or remdesivir (S1 and S2 Tables in S1 File). Mortality between the patients who received at least one of these interventions/studies was statistically similar to those who were not subject to open label interventions or trial enrollment, with 2 deaths in the group of 60 patients enrolled in a trial or receiving open label therapy versus 5 deaths out of the 52 who did not (p = 0.171 by chi squared, unadjusted analysis). The correlation of any individual therapy with mortality could not be determined due to the overall low mortality rate of our cohort.

Treatment guidance at our institution recommended the use of pharmacologic VTE prophylaxis on all patients unless contraindicated. Therapeutic anticoagulation for VTE was recommended only in the case of confirmed or highly suspected VTE. In our cohort, 26/112 received therapeutic dose anticoagulation; 25 of these patients (96%) had a known indication for therapeutic anticoagulation, with only 1 patient receiving anticoagulation empirically. Indications for

therapeutic anticoagulation included chronic conditions (atrial fibrillation, bioprosthetic valve or prior VTE, totaling 7 patients), new atrial fibrillation (6 patients), and need for extracorporeal support (1 patient). Eleven patients had suspected or confirmed acute thromboembolic events during their hospitalization, including deep venous thrombosis or pulmonary embolus (6 patients), frequent clotting of dialysis catheters (2 patients), acute coronary syndrome (1 patients), ischemic digit (1 patient), or left ventricular thrombus (1 patient) (S3 Table in S1 File). Of the 6 patients (5.4%) who developed radiographically-confirmed VTE, 2 were diagnosed at the time of admission. Of the 4 who later developed VTE, all were on VTE prophylaxis at the time of diagnosis. Three of the patients with radiographically-confirmed VTE had ARDS. None of the patients with new VTE died during their hospitalization. Mortality was not significantly different amongst those receiving and not receiving therapeutic anticoagulation (4% v. 7%, $\chi^2$ p = 0.563).

While there has been debate about the use of steroids in the treatment of COVID-19 at the time of data collection, our institutional treatment guidance did not recommend routine steroids for COVID-19 related ARDS in the absence of another indication during this time period. From our cohort, 11/112 received steroid doses that exceeded 20 mg of prednisone equivalents daily. Of those, 10/11 had an indication based on past medical history (e.g., chronic steroid use), new adrenal insufficiency, or refractory shock.

### ARDS management

In patients who developed ARDS, we examined the rate of adherence to lung protective ventilation and the frequency of adjunctive therapy use. Patients with ARDS had a median tidal volume of 6.1 cc/kg PBW on day 1. The pooled median tidal volumes per PBW in days 2–7 varied from 5.9 to 6.1 (Fig 2). Outliers with larger tidal volumes tended to be those patients placed on pressure support. Adherence to LPV (<8cc/kg PBW) was 90.2% for ventilator days 1–7. We also evaluated the use of PEEP in our cohort. For patients with mild to moderate ARDS (P/F > 150, n = 3), median PEEP was between 5–10 cmH$_2$O depending on the day for the first 7 days of mechanical ventilation. For moderate to severe ARDS (P/F ratio < 150, n = 21), median daily PEEP was 10–12 cmH$_2$O for the first 7 days of mechanical ventilation.

Out of 21 patients with ARDS and P/F ratio < 150, 12 patients (57%) underwent prone positioning at least once. Among those with available data (n = 9), the P/F ratio improved by an average of 63 points (SD 84) after the first proning event. In addition, out of the 28 intubated patients, 14/28 (50%) received neuromuscular blockade outside of rapid sequence intubation. For the intubated patients, compliance on day of intubation (day 1) had a median value of 34.1 ml/cm H$_2$O (IQR = 24.6–41.8, mean 34.0, SD 10.4), for day 3 the median was 30.3 (IQR 23.9–40.1, mean 32.7, SD 10.4), for day 5 the median was 34.6 (IQR 26.9–39.9, mean 35.4, SD 12.4) and for day 7 the median was 34.7 (IQR 30.7–52.5, mean 38.8, SD 11.7). Only a minority (2/28) of intubated patients received inhaled nitric oxide at any point, and only one patient received veno-venous extracorporeal membrane oxygenation (VV-ECMO).

## Discussion

In this retrospective, observational study, we report lower hospital mortality in both critically and non-critically ill patients at our center in comparison with prior published cohorts. Patient-level characteristics associated with need for invasive mechanical ventilation included obesity, past and present smoking history, and elevation of pro-inflammatory markers. Further understanding of the factors associated with clinical deterioration may help identify therapeutic targets for early intervention.

To our knowledge, this is the first study identifying obesity and smoking history as risk factors for the need for invasive mechanical ventilation in COVID-19. While obesity has been

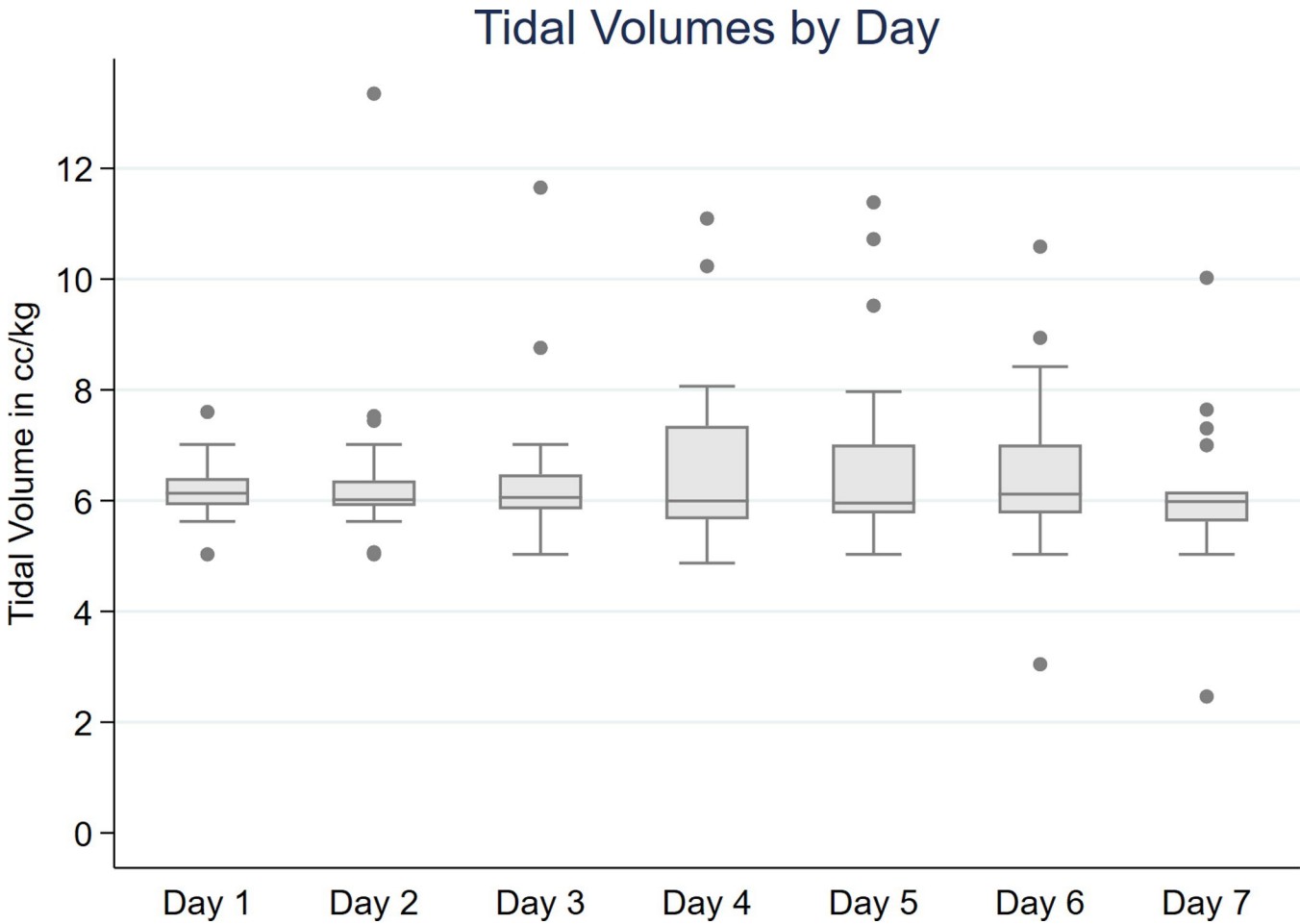

**Fig 2. Box and whisker plot of pooled cohort mechanical ventilation tidal volume for first 7 days post-intubation.** Tidal volume is displayed as cc/kg of predicted body weight. Box represents IQR and whiskers represent minimum and maximum, with outliers represented as dots.

associated with increased risk for a positive COVID-19 test [11], and an association between obesity and mortality in COVID-19 has been speculated [12,13], its association with respiratory outcomes in COVID-19 has not been directly evaluated. On the other hand, the impact of smoking in COVID-19 disease severity may be explained by the observation that smokers have upregulation of ACE2 receptors on lung biopsy [14,15]. Indeed, a large observational cohort from Wuhan reported a higher rate of smoking among patients with more severe forms of COVID-19 [16]. In addition, a smaller study revealed a higher adjusted risk for disease progression in patients with a history of smoking compared to never smokers [17]. Metanalyses evaluating the relationship between smoking and COVID-19 disease severity have had mixed results. A smaller evaluation of five published cohorts from China did not find a statistically significant association between active smoking and disease severity; however the analysis was unadjusted [18]. A larger metanalysis found that current or former smoking was associated higher unadjusted risk for the composite outcome of ICU admission, intubation or death in COVID-19 [19]. None of the included studies examined adjusted risk for IMV [19,20].

Overall mortality in our patient population was low. While the factors associated with mortality remain unclear, this finding is encouraging. This cohort shared some similarities to others, including a majority of male patients (66%) [16,21–24], a median age of 61 [21,23,24], and an admission rate of 56% non-white minorities [24]. We also report comparable rates of

invasive mechanical ventilation to other cohorts [21,24]. However, the role of baseline severity of illness was not fully assessed, and has not been extensively detailed in prior studies. As such, it is possible that our patient population was less ill than others.

In our ventilated patients, the in-hospital mortality rate was 18%, which was similar to the rate seen in a recently published study, but lower than that seen in earlier cohorts [23–25]. The vast majority of mechanically ventilated patients in our cohort had ARDS, suggesting that optimizing ARDS care may improve the outcomes of these patients. ARDS is traditionally under-diagnosed worldwide, and prior studies demonstrate the underutilization of therapies known to reduce mortality in this syndrome [26,27].

Both lung protective ventilation and prone positioning have been associated with improved outcomes in ARDS and are recommended by multiple professional society guidelines [6,7,28,29]. Our adherence to lung protective ventilation strategies (90.2% for ventilator days 1–7) was higher than that reported in previous large multicenter ARDS cohorts [27,30]. Prone positioning was also used almost 60% of the time compared with 8–11% in prior reports [26,27]. While we were unable to definitively correlate any individual therapeutic measure with mortality given the small size of our cohort, adherence to established ARDS best practices may have played a role. It is also likely that the lower per capita incidence of COVID-19 in our state [31] and the lack of strain in our system enabled adherence to these labor-intensive interventions for ARDS. Indeed, as of the time of this writing, there was no change to usual staffing ratios, or need for non-ICU clinicians to take care of ICU level patients. Although this finding warrants further investigation, it may suggest that efforts to mitigate the spread of COVID-19 as a means to avoid straining healthcare systems may indeed be an effective strategy for controlling mortality from this disease.

Regarding features that may be specific to COVID-19, there have been several reports of high rates of coagulopathy and VTE in these patients [32–34]. Post-mortem studies revealed a high rate of microthrombi in the pulmonary vasculature of COVID-19 patients [35]. Microvascular damage and thrombi formation are in fact well documented in ARDS [36,37]. We found a low rate of VTE (5%) in our patient population. While VTE was not a focus of our study, some mitigating factors associated with its seemingly low incidence in our cohort may include a high use of VTE prophylaxis, as well as recommendations by our pandemic response team for daily awakenings, light sedation, and regular consultation with physical and occupational therapy.

Recent reports have also suggested that COVID-19 produces an excessive inflammatory response by the host. Indeed, IL-6 and serum ferritin, traditionally elevated in proinflammatory conditions, have been associated with higher mortality in patients infected with COVID-19 [16]. In this cohort, ferritin and IL-6 also correlated with increased rates of endotracheal intubation, but no definitive correlation with mortality could be made. Given the concern that excessive inflammatory response may play a role in mortality, multiple clinical trials have been initiated to inhibit a number of targets including interleukin 1, interleukin 6, and granulocyte-macrophage-colony stimulating factor. While a large portion of our patients were enrolled in clinical trials, the impact of these agents remains to be seen.

The strengths of this study included manual extraction of all data points, which allowed for higher accuracy and granularity of findings. In addition, the in-hospital mortality was known for all of our patients. Study weaknesses included the single center nature and small sample size, which precluded more robust evaluation of risk factors associated with mortality. In addition, short follow up times precluded the assessment of disease sequela in our cohort. We also did not collect information on social determinants of health such as education level, insurance status or economic factors, all of which could have influenced observed outcomes. Finally, given its retrospective nature, we also cannot ascertain that any of our interventions directly affected patient outcomes. Nevertheless, our data identified potential risk factors for disease progression.

## Conclusion

Early experience with COVID-19 at UCLA has revealed a lower mortality than previously reported. Baseline factors associated with increased odds for the need for mechanical ventilation include obesity and smoking history; further research is needed to confirm these findings in larger, multicenter cohorts. Finally, the incidence of ARDS among intubated patients with COVID-19 is high, suggesting that optimizing ARDS care may improve the outcomes of these patients.

## Supporting information

**S1 File.**
(PDF)

## Author Contributions

**Conceptualization:** Ana C. Monteiro, Rajat Suri, Iheanacho O. Emeruwa, Robert J. Stretch, Alexander Sherman, Tisha Wang, Nida Qadir.

**Data curation:** Ana C. Monteiro, Rajat Suri, Robert J. Stretch, Roxana Y. Cortes-Lopez, Alexander Sherman, Catherine C. Lindsay, Jennifer A. Fulcher, David Goodman-Meza.

**Formal analysis:** Ana C. Monteiro, Rajat Suri, Robert J. Stretch, Alexander Sherman, Russell G. Buhr.

**Funding acquisition:** Ana C. Monteiro.

**Investigation:** Ana C. Monteiro, Rajat Suri, Robert J. Stretch, Jennifer A. Fulcher, David Goodman-Meza.

**Methodology:** Ana C. Monteiro, Rajat Suri, Robert J. Stretch, Alexander Sherman, Tisha Wang, Nida Qadir.

**Project administration:** Ana C. Monteiro, Iheanacho O. Emeruwa, Robert J. Stretch, Nida Qadir.

**Resources:** Ana C. Monteiro.

**Software:** Rajat Suri, Robert J. Stretch.

**Supervision:** Anil Sapru, Russell G. Buhr, Steven Y. Chang, Tisha Wang, Nida Qadir.

**Validation:** Russell G. Buhr, Steven Y. Chang, Tisha Wang.

**Visualization:** Ana C. Monteiro.

**Writing – original draft:** Ana C. Monteiro, Rajat Suri, Roxana Y. Cortes-Lopez.

**Writing – review & editing:** Ana C. Monteiro, Rajat Suri, Iheanacho O. Emeruwa, Jennifer A. Fulcher, David Goodman-Meza, Russell G. Buhr, Steven Y. Chang, Tisha Wang, Nida Qadir.

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
