## [Decision Letter · Decision Letter 0]

3 Oct 2020

PONE-D-20-26498

Obesity and Smoking as Risk Factors for Invasive Mechanical Ventilation in COVID-19: a Retrospective, Observational Cohort Study

PLOS ONE

Dear Dr. Costa Monteiro,

Thank you for submitting your manuscript to PLOS ONE. After careful consideration, we feel that it has merit but does not fully meet PLOS ONE’s publication criteria as it currently stands. Therefore, we invite you to submit a revised version of the manuscript that addresses the points raised during the review process.

We look forward to receiving your revised manuscript.

Kind regards,

Giordano Madeddu

Academic Editor

PLOS ONE

Journal Requirements:

2. Please include the date(s) on which you accessed the databases or records to obtain the data used in your study.

3. Please define current, former, and never smoker statuses.

"I have read the journal's policy and the authors of this manuscript have the following competing interests: SYC consults for PureTech on their deupirfenidone in COVID study. "

Reviewers' comments:

Reviewer's Responses to Questions

**Comments to the Author**

1. Is the manuscript technically sound, and do the data support the conclusions?

Reviewer #1: Yes

Reviewer #2: Yes

2. Has the statistical analysis been performed appropriately and rigorously? 

Reviewer #1: I Don't Know

Reviewer #2: Yes

3. Have the authors made all data underlying the findings in their manuscript fully available?

Reviewer #1: Yes

Reviewer #2: Yes

4. Is the manuscript presented in an intelligible fashion and written in standard English?

Reviewer #1: Yes

Reviewer #2: Yes

5. Review Comments to the Author

Reviewer #1: The goal of this study was to evaluate respiratory failure in COVID-19 and explore factors associated

with risk of invasive mechanical ventilation (IMV). A retrospective, observational cohort study of 112 inpatient adults diagnosed with COVID-19 was carried out. Data were manually extracted from electronic

medical records. Multivariable and Univariable regression were used to evaluate association between

baseline characteristics, initial serum markers and the outcome of IMV. It was determined that obesity, smoking history, and elevated inflammatory markers were associated with increased need for IMV in patients with COVID-19. These findings are of utmost importance given the current scenario.

Reviewer #2: In this study (PONE-D-20-26498), Monteiro and colleagues described a small sample (28), who needed endotracheal intubation and invasive mechanical ventilation, among 112 COVID-19 included in the analysis. The idea is interesting the paper is well written, however I have some considerations to explain. First of all they described that 28 patients out 47 admitted to the ICU were intubated, but they did not describe the criteria of intubation.

Could you describe the intubation criteria that you used, please?

Moreover they did not describe any severity score, i.e. APACHE II, SAPS II.

Could you add a severity score to understand better the severity of your population admitted to ICU, since your mortality rate was very low compared to other ICU?

Regarding mechanical ventilation, they described a setting of median PEEP level of 10-12 cmH2O in severe ARDS and P/F rati < 150, but they did not describe any data on plateau pressure, driving pressure or compliance of the respiratory system. Therefore how did you decide to set PEEP level? Could you you described better respiratory mechanics (plateau pressure, driving pressure, PEEPi, compliance of the respiratory system)? Which criteria did you consider to decide for prone positioning? Could you report the duration of mechanical ventilation?

6. PLOS authors have the option to publish the peer review history of their article (what does this mean?). If published, this will include your full peer review and any attached files.

Reviewer #1: No

Reviewer #2: **Yes: **Daniela Pasero

---

## [Author Response · Author response to Decision Letter 0]

29 Oct 2020

This Document has also been uploaded for ease of viewing. 

We have revised the manuscript to comply to the above guidelines. 

2. Please include the date(s) on which you accessed the databases or records to obtain the data used in your study.

We have added the following sentence to the manuscript methods section: 

“This is a retrospective observational cohort study conducted from March 12 2020 and April 16 2020 approved by the UCLA (University of California, Los Angeles) institutional review board with waiver of informed consent. We evaluated the first 113 unique admissions to our healthcare system. We originally accessed patient charts between March 12 and June 16th 2020. We returned to the charts in October 2020 to extract elements requested during the review process.”

3. Please define current, former, and never smoker statuses.

We have added these definitions to the methods section: 

“We extracted smoking status from the admission note. Patients were interviewed about smoking history on admission. Never smokers, prior smokers and current smokers were self-identified as such by the patients during the interview. Those patients who self-identified as prior smokers were then asked when they last smoked and number of packs a day. Those who identified as current smokers were asked the number of packs smoked a day. Patients who smoked less than a pack-year were considered never smokers.”

4. Thank you for stating the following in the Competing Interests section: "I have read the journal's policy and the authors of this manuscript have the following competing interests: SYC consults for PureTech on their deupirfenidone in COVID study. "

 We have added the following to our cover letter: 

“I have read the journal's policy and the authors of this manuscript have the following competing interests: SYC consults for PureTech on their deupirfenidone in COVID study. This does not alter our adherence to PLOS ONE policies on sharing data and materials.”

 We have since uploaded the data used in the manuscript in the public repository https://datadryad.org/stash. The DOI is https://doi.org/10.5068/D1QX18. It will be made public once the manuscript is accepted. In the meantime you may access this at https://datadryad.org/stash/share/SJtUG1lXSBD9KlQcQapB1x1g7yJAvhFxqcInaILQy1w

Thank you, we have completed this. 

In Response to reviewers' comments:

Reviewer #1: The goal of this study was to evaluate respiratory failure in COVID-19 and explore factors associated with risk of invasive mechanical ventilation (IMV). A retrospective, observational cohort study of 112 inpatient adults diagnosed with COVID-19 was carried out. Data were manually extracted from electronic medical records. Multivariable and Univariable regression were used to evaluate association between baseline characteristics, initial serum markers and the outcome of IMV. It was determined that obesity, smoking history, and elevated inflammatory markers were associated with increased need for IMV in patients with COVID-19. These findings are of utmost importance given the current scenario.

We appreciate your support of our manuscript. 

Reviewer #2: In this study (PONE-D-20-26498), Monteiro and colleagues described a small sample (28), who needed endotracheal intubation and invasive mechanical ventilation, among 112 COVID-19 included in the analysis. The idea is interesting the paper is well written, however I have some considerations to explain. First of all they described that 28 patients out 47 admitted to the ICU were intubated, but they did not describe the criteria of intubation. Could you describe the intubation criteria that you used, please?

Thank you for pointing this out. In the methods we have now added the following statement: 

“Guidelines recommended intubation for patients who had PCR confirmed COVID-19 and who demonstrated rapid escalation of oxygen requirements. It was at the discretion of the clinician to decide whether a patient would tolerate a trial of non-invasive oxygen delivery via non-rebreather mask or high flow nasal canula (HFNC). If the clinician decided that trial of these non-invasive therapies would place the patient and/or staff at greater risk, the patient would proceed to intubation.”

- Moreover they did not describe any severity score, i.e. APACHE II, SAPS II. Could you add a severity score to understand better the severity of your population admitted to ICU, since your mortality rate was very low compared to other ICU?

We appreciate the suggestion. We re-evaluated the COVID literature and found a recent study on COVID-19 patients from Wuhan which reported that APACHE II outperformed SOFA and CURB-65 as a predictor of mortality in this population (https://www.ncbi.nlm.nih.gov/pmc/articles/PMC7217128/). As such, we chose to calculate APACHE II for the patients in our cohort who were admitted to the ICU. We have added the following to the results sections: 

“Out of the 47 ICU admissions, the median APACHE II score on arrival to the ICU was 12 (IQR 7-16, mean 12.7, SD 7.11).” 

In comparison, the cohort from Wuhan, China had a mean APACHE score of 15 +- SD 7.7 (https://www.ncbi.nlm.nih.gov/pmc/articles/PMC7217128/). Considering the population size, standard deviation and means of these two populations, a two tailed t-test revealed that the APACHE scores in these two populations were not significantly different. 

- Regarding mechanical ventilation, they described a setting of median PEEP level of 10-12 cmH2O in severe ARDS and P/F ratio < 150, but they did not describe any data on plateau pressure, driving pressure or compliance of the respiratory system. Therefore how did you decide to set PEEP level? Could you describe better respiratory mechanics (plateau pressure, driving pressure, PEEP, compliance of the respiratory system)? Which criteria did you consider to decide for prone positioning? Could you report the duration of mechanical ventilation?

As stated in our methods, we recommended PEEP titration per ARDS network PEEP tables and prone positioning when P/F <150, as recommended by the PROSEVA trial: “…early consideration of prone ventilation for patients with P/F ratios < 150” and “Titration of positive end-expiratory pressure (PEEP) was recommended to reflect ARDS network PEEP tables”. 

We have also added that we recommended PPlat to stay under 30 cm H2O. 

We have since included the static compliance of patients who were intubated, as measured as part of our respiratory therapy routine care. We have added the following to our manuscript: 

“ For the intubated patients, compliance on day of intubation (day 1) had a median value of 34.1 ml/cm H2O (IQR = 24.6-41.8, mean 34.0, SD 10.4), for day 3 the median was 30.3 (IQR 23.9-40.1, mean 32.7, SD 10.4), for day 5 the median was 34.6 (IQR 26.9-39.9, mean 35.4, SD 12.4) and for day 7 the median was 34.7 (IQR 30.7-52.5, mean 38.8, SD 11.7).”

Interestingly, the static compliance measured in these patients were most similar to the non COVID ARDS arm reported in the Grasselli study (https://www.thelancet.com/journals/lanres/article/PIIS2213-2600%2820%2930370-2/fulltext), in which they reported a median compliance of 32 ml/cm H2O (IQR 25-43), and lower than the median of 41 reported for the COVID-19 arm of the same study.

---

## [Decision Letter · Decision Letter 1]

8 Dec 2020

Obesity and Smoking as Risk Factors for Invasive Mechanical Ventilation in COVID-19: a Retrospective, Observational Cohort Study

PONE-D-20-26498R1

Dear Dr. Costa Monteiro,

We’re pleased to inform you that your manuscript has been judged scientifically suitable for publication and will be formally accepted for publication once it meets all outstanding technical requirements.

Kind regards,

Giordano Madeddu

Academic Editor

PLOS ONE

Additional Editor Comments (optional):

Reviewers' comments:

Reviewer's Responses to Questions

**Comments to the Author**

1. If the authors have adequately addressed your comments raised in a previous round of review and you feel that this manuscript is now acceptable for publication, you may indicate that here to bypass the “Comments to the Author” section, enter your conflict of interest statement in the “Confidential to Editor” section, and submit your "Accept" recommendation.

Reviewer #1: All comments have been addressed

Reviewer #2: All comments have been addressed

2. Is the manuscript technically sound, and do the data support the conclusions?

Reviewer #1: Yes

Reviewer #2: Yes

3. Has the statistical analysis been performed appropriately and rigorously? 

Reviewer #1: Yes

Reviewer #2: Yes

4. Have the authors made all data underlying the findings in their manuscript fully available?

Reviewer #1: Yes

Reviewer #2: Yes

5. Is the manuscript presented in an intelligible fashion and written in standard English?

Reviewer #1: Yes

Reviewer #2: Yes

6. Review Comments to the Author

Reviewer #1: (No Response)

Reviewer #2: Congratulations,

the authors addressed all comments, and the manuscript is now suitable for acceptance.

7. PLOS authors have the option to publish the peer review history of their article (what does this mean?). If published, this will include your full peer review and any attached files.

Reviewer #1: **Yes: **Shama Ahmad

Reviewer #2: **Yes: **Daniela Pasero

---

## [Editor Report · Acceptance letter]

10 Dec 2020

PONE-D-20-26498R1 

Obesity and smoking as risk factors for invasive mechanical ventilation in COVID-19: A retrospective, observational cohort Study 

Dear Dr. Monteiro:

I'm pleased to inform you that your manuscript has been deemed suitable for publication in PLOS ONE. Congratulations! Your manuscript is now with our production department. 

Kind regards, 

on behalf of

Dr. Giordano Madeddu 

Academic Editor

PLOS ONE